# A Comparison of the Correlation of Systemic Lupus Erythematosus Disease Activity Index 2000 (SLEDAI-2K) and Systemic Lupus Erythematosus Disease Activity Score (SLE-DAS) with Health-Related Quality of Life

**DOI:** 10.3390/jcm10102137

**Published:** 2021-05-15

**Authors:** Ning-Sheng Lai, Ming-Chi Lu, Hsiu-Hua Chang, Hui-Chin Lo, Chia-Wen Hsu, Kuang-Yung Huang, Chien-Hsueh Tung, Bao-Bao Hsu, Cheng-Han Wu, Malcolm Koo

**Affiliations:** 1Division of Allergy, Immunology and Rheumatology, Dalin Tzu Chi Hospital, Buddhist Tzu Chi Medical Foundation, Dalin, Chiayi 622401, Taiwan; tzuchilai@gmail.com (N.-S.L.); e360187@yahoo.com.tw (M.-C.L.); hky0919@yahoo.com.tw (K.-Y.H.); dr5188@yahoo.com.tw (C.-H.T.); baobaohsu100@yahoo.com.tw (B.-B.H.); wickham_wu@yahoo.com.tw (C.-H.W.); 2School of Medicine, Tzu Chi University, Hualien City, Hualien 97004, Taiwan; 3Department of Medical Research, Dalin Tzu Chi Hospital, Buddhist Tzu Chi Medical Foundation, Dalin, Chiayi 622401, Taiwan; df274760@tzuchi.com.tw (H.-H.C.); df289469@tzuchi.com.tw (H.-C.L.); chiawen0114@yahoo.com.tw (C.-W.H.); 4Graduate Institute of Long-Term Care, Tzu Chi University of Science and Technology, Hualien City, Hualien 973302, Taiwan; 5Dalla Lana School of Public Health, University of Toronto, Toronto, ON M5T 3M7, Canada

**Keywords:** systemic lupus erythematosus, quality of life, cross-sectional studies, surveys and questionnaires

## Abstract

Background and Aim: The aim of this study was to compare the correlation of a recently developed systemic lupus erythematosus disease activity score (SLE-DAS) with the SLE disease activity index 2000 (SLEDAI-2K) with the Lupus Quality of Life questionnaire (LupusQoL) in Taiwanese patients with SLE. Methods: A cross-sectional study was conducted in a regional teaching hospital in Taiwan from April to August 2019. Adult patients with a clinician-confirmed diagnosis of SLE based on the 1997 American College of Rheumatology revised criteria or the 2012 Systemic Lupus International Collaborating Clinics Classification Criteria were recruited. SLE disease activity was measured with both SLEDAI-2K and SLE-DAS. Disease-specific quality of life was assessed using the LupusQoL. Results: Of the 333 patients with SLE in this study, 90.4% were female and 40% were between the ages of 20 and 39 years. The median SLEDAI-2K score was 4.00 (interquartile range [IQR] 2.00–7.50) and the median SLE-DAS score was 2.08 (IQR 1.12–8.24) in our patients with SLE. After adjusting for sex and age intervals, both SLEDAI-2k and SLE-DAS were significantly and inversely associated with all eight domains of LupusQoL. The magnitudes of the mean absolute error, root mean square error, Akaike Information Criterion, Bayesian Information Criterion, and coefficient of determination were comparable between SLEDAI-2K and SLE-DAS. Conclusions: There were no clear differences in the use of SLE-DAS over SLEDAI-2K in assessing HRQoL in patients with SLE. We suggest that, in this aspect, both SLEDAI-2K and SLE-DAS are effective tools for measuring disease activity in patients with SLE.

## 1. Introduction

Systemic lupus erythematosus (SLE) is a chronic systemic autoimmune disease involving multiple organ systems, such as the skin, kidneys, blood, joints, and brain [1]. The disease predominantly affects women of childbearing age, with female-to-male ratio of 9 to 1. The clinical course of SLE is highly variable with recurrent relapses and exacerbations. Despite the advancements in therapeutic options and the improvement in the survival rate for SLE [2], a high proportion of patients living with SLE have a poor health-related quality of life (HRQoL) compared with healthy individuals as well as patients with other chronic diseases, such as diabetes, hypertension, and even heart failure [3]. Fatigue, pain, and musculoskeletal distress associated with SLE have been reported to be the main predictors of poor HRQoL [4]. Older age, poverty, lower educational level, behavioral issues, some clinical manifestations, and comorbidities could also have an impact on HRQoL [5]. In addition, disease activity status has been suggested to adversely affect HRQoL in patients with SLE [6,7,8,9,10,11].

One of the most commonly used measures for the global disease activity of SLE is the Systemic Lupus Erythematosus Disease Activity Index 2000 (SLEDAI-2K) introduced in 2002. It is a modification of the original Systemic Lupus Erythematosus Disease Activity Index (SLEDAI) developed by consensus of a group of experienced clinicians in the field of lupus research [12]. The SLEDAI-2K was validated against SLEDAI in a cohort of 960 patients and a high correlation of 0.97 between the two indices was reported [13]. More recently, a new 17-item Systemic Lupus Erythematosus Disease Activity Score (SLE-DAS) with improved sensitivity to changes in SLE disease activity as compared with SLEDAI was proposed. In a study of 520 patients with SLE, the SLE-DAS showed a significantly better performance than SLEDAI-2K in identifying clinically meaningful changes in disease activity and in predicting damage accrual [14]. The scale was subsequently validated in an independent cohort of 227 Latin American patients with Mexican Mestizo ethnicity. Nevertheless, the authors concluded that SLE-DAS did not add an advantage over the existing SLEDAI-2K score, particularly regarding its suboptimal performance in patients with high disease activity [15]. In addition, the choice of outcome measures for the musculoskeletal component in SLE-DAS has been challenged by a study that reanalyzed the data with SLE-DAS obtained from a longitudinal study of patients with SLE [16]. Furthermore, another study retrospectively calculated SLE-DAS for 41 patients with lupus nephritis and revealed that the performance of SLE-DAS among patients of high disease activity might not be robust. The authors concluded that there might be no added advantage over the existing SLEDAI-2K score in the current state of SLE-DAS [17].

Given that measuring SLE disease activity remains a challenging and complex task, it is clear that a broader evaluation of the new SLE-DAS is needed, particularly, in diverse populations across a spectrum of severity and types of clinical manifestations of SLE [18]. At present, no studies have yet attempted to compare the correlation of these two indices in predicting HRQoL in patients with SLE. Therefore, the aim of this cross-sectional study was to compare the correlation of SLEDAI-2K and SLE-DAS with a disease-specific HRQoL, the Lupus Quality of Life questionnaire (LupusQoL) [19], in patients with SLE.

## 2. Materials and Methods

### 2.1. Study Design and Study Population

This cross-sectional study was conducted at the rheumatology outpatient clinic in a regional hospital in southern Taiwan from April to August 2019. Patients were consecutively enrolled and all participants signed informed consent under a study protocol approved by the institutional review board of the Dalin Tzu Chi Hospital, Buddhist Tzu Chi Medical Foundation (No. B10801017). The study was carried out in accordance with the Declaration of Helsinki.

Patients aged 20 years and older, with a clinician-confirmed diagnosis of SLE based on the 1997 American College of Rheumatology (ACR) revised criteria [20] or the 2012 Systemic Lupus International Collaborating Clinics Classification Criteria [21] were recruited. The exclusion criteria included patients who had previously been diagnosed with other major systemic diseases, including rheumatoid arthritis, polymyositis, dermatomyositis, systemic sclerosis, spondyloarthritis, and juvenile idiopathic arthritis.

### 2.2. Measurement of Disease Activity

SLE disease activity was assessed using rheumatologist-scored SLEDAI-2K [13] and SLE-DAS [14]. The SLEDAI-2K consists of 24 items covering nine organ systems. The recall period for disease activity is the previous 10 days. The score ranges from 0 to 105 points, with higher values signifying greater disease activity.

The SLE-DAS consists of 17 items and has all disease manifestations in the 24-item SLEDAI-2K with added items for hemolytic anemia, cardiopulmonary, and gastrointestinal involvement. The SLE-DAS is a continuous disease activity score with higher values signifying greater disease activity [14].

### 2.3. Measurement of Disease-Specific Quality of Life and Other Variables

Demographic and clinical information of the patients was collected using a paper-based questionnaire consisting of questions on sex, age interval, body mass index, educational level, marital status, job change due to SLE, employment status, self-perceived health status, duration of SLE, age of diagnosis of SLE, alcohol use, smoking, betel nut chewing, regular exercise, and sleep duration. The questionnaire was administered by two experienced research nurses of the rheumatology outpatient clinic.

The LupusQoL, which is one of the most validated measures of disease-specific HRQoL in patients with SLE, was used in this study [22]. The original LupusQoL was developed from qualitative interviews with patients with SLE and expert panel agreement followed by psychometric evaluation [19]. The LupusQoL consists of 34 items grouped in eight domains of HRQoL, including physical health (8 items), emotional health (6 items), body image (5 items), pain (3 items), planning (3 items), fatigue (4 items), intimate relationships (2 items), and burden to others (3 items). The recall period is the previous four weeks. The response scale was a five-point Likert format, where 0 = all of the time, 1 = most of the time, 2 = a good bit of the time, 3 = occasionally, and 4 = never. For each domain, the mean domain score is obtained by dividing the total score by the number of items in that domain. The mean domain score is rescaled to a final score ranging from 0 to 100 by dividing by 4 (the number of Likert responses minus 1) and then multiplying by 100. A non-applicable response is available in six of the items, and it is treated as unanswered. A higher score in a domain indicates a better health-related quality of life for that particular domain. The validity of the original English version of LupusQoL has been demonstrated in patients with SLE in the United Kingdom [19] and the United States [23]. In this study, we used the official Chinese for Taiwan version of the LupusQoL, which was obtained from RWS Life Sciences with permission for use in this study. A study in China on 208 patients with SLE, using the LupusQoL-China culturally adapted from the Chinese for Taiwan version, demonstrated evidence of construct validity when compared with equivalent domains on the EQ-5D. In addition, the internal consistency reliability Cronbach’s α ranged from 0.81 to 0.96 with the test–retest reliability ranging from 0.84 to 0.97 across the different domains for the LupusQoL-China [24].

### 2.4. Statistical Analysis

All statistical analyses were performed using SAS software release 9.4 (SAS Institute, Inc., Cary, NC, USA). Continuous variables were summarized as mean with standard deviation (SD) and median with interquartile range, as appropriate. Categorical variables were presented as frequencies and percentages. Separate linear regression analyses for each of the eight domains of LupusQoL were performed with SLEDAI-2K and SLE-DAS as independent variables. Because sex differences were observed in HRQoL in patients with SLE [25], linear regression models were fitted with and without adjusting for sex and age interval.

The correlations of SLEDAI-2K and SLE-DAS with LupusQoL were assessed using five regression model accuracy metrics, including mean absolute error (MAE), root mean square error (RMSE), Akaike Information Criterion (AIC), Bayesian Information Criterion (BIC), and coefficient of determination (R^2^). The MAE is the average of the absolute differences between prediction and actual observation with all individual differences has equal weight. The RMSE is the square root of the average of squared differences between prediction and actual observation, and therefore it gives relatively high weight to large errors. A smaller value in MAE and RMSE indicates better model performance. Similarly, a lower AIC or BIC value indicates a better model fit. Conversely, because R^2^ is the proportion of variation in the outcome that is explained by the predictor variables; therefore the higher the R^2^, the better the model [26]. The differences in the MAE and RMSE between SLEDAI-2K and SLE-DAS were compared using the paired *t*-test. In addition, the correlation between SLEDAI-2K and SLE-DAS was determined using Pearson’s correlation coefficient and Spearman’s rank correlation coefficient.

## 3. Results

The demographic and clinical information of the 333 patients with SLE are shown in Table 1. In brief, 90.4% were female and 40% were between the ages of 20 and 39 years. Approximately 54% of the patients had a normal body mass index, and 50% had an educational level of college or above. About 29% had to change their jobs due to SLE, and 73% rated their own health as average or below. In addition, 64% of the patients had SLE for more than nine years. In addition, 55.3% patients with SLE had low complement levels and 35.1% had increased anti-double strain DNA antibody titer. Clinically, 61.6% patients with SLE had Raynaud’s phenomenon and 51.7% had photosensitivity.

Summary statistics of SLEDAI-2K, SLE-DAS, and individual domains of LupusQoL are also presented in Table 2. The median SLEDAI-2K and SLE-DAS was 4.00 (interquartile range [IQR] 2.00–7.50) and 2.08 (IQR 1.12–8.24), respectively. Figure 1 shows a scatter plot of SLEDAI-2K and SLE-DAS. There was a moderate correlation between SLEDAI-2K and SLE-DAS (Pearson’s r = 0.66; 95% CI 0.60, 0.72; *p* < 0.001; Spearman’s ρ = 0.78; 95% CI 0.71, 0.83; *p* < 0.001).

Table 3 and Table 4 show the association of the eight domains of LupusQoL with SLEDAI-2K and SLE-DAS, respectively. In Table 3, SLEDAI-2K was significantly and inversely associated with five domains of LupusQoL, namely, emotional health (*p* = 0.036), body image (*p* = 0.033), pain (*p* = 0.033), fatigue (*p* = 0.003), and burden to others (*p* < 0.001). When adjusting for sex and age interval, SLEDAI-2K became significantly and inversely associated with all eight domains of LupusQoL. The standardized beta coefficients for the eight domains ranged from the highest at −0.238 in burden to others to the lowest at −0.123 in planning. The three domains with the highest standardized beta coefficients were burden to others (−0.238), followed by pain (−0.196) and physical health (−0.192).

In Table 4, SLE-DAS was significantly and inversely associated with six domains of LupusQoL, namely, physical health (*p* = 0.003), emotional health (*p* = 0.007), pain (*p* = 0.002), fatigue (*p* = 0.001), intimate relationships (*p* = 0.022), and burden to others (*p* < 0.001). When adjusting for sex and age interval, SLE-DAS also became significantly and inversely associated with all eight domains of LupusQoL. The standardized beta coefficients for the eight domains ranged from the highest at −0.217 in physical health to the two lowest at −0.115 in planning and body image. The three domains with the highest standardized beta coefficients were physical health (−0.217), followed by burden to others (−0.216), and pain (−0.203).

Correlations of SLEDAI-2K and SLE-DAS with LupusQoL were evaluated by comparing five regression model accuracy metrics (Table 5). The magnitudes of MAE, RMSE, AIC, BIC, and R^2^ were comparable between SLEDAI-2K and SLE-DAS. In addition, MAE and RMSE obtained from SLEDAI-2K and SLE-DAS were not significantly different for all eight domains of LupusQoL.

In Table 6 and Table 7, correlations of SLEDAI-2K and SLE-DAS with LupusQoL in patients with or without renal involvement were evaluated by comparing five regression model accuracy metrics. The magnitudes of MAE, RMSE, AIC, BIC, and R^2^ were comparable between SLEDAI-2K and SLE-DAS. In addition, MAE and RMSE obtained from SLEDAI-2K and SLE-DAS were not significantly different for all eight domains of LupusQoL in patients with SLE with renal involvement or not.

## 4. Discussion

Measuring disease activity in patients with SLE is important but complex. In this study on 333 patients with SLE, a commonly used SLEDAI-2K was compared with a more recently developed SLE-DAS scoring tool. Overall, we found that the correlations between SLEDAI-2K and SLE-DAS with HRQoL, as measured by LupusQoL, were similar in our patients with SLE. We used five regression model accuracy metrics to assess the performance of the two disease activity measures, and no clear advantages were observed with the newer SLE-DAS over the SLEDAI-2K with respect to their associations with HRQoL. In addition, while there were small differences in the magnitude of the R^2^ between the SLEDAI-2K and SLE-DAS, the differences were not in the same direction for the eight domains of LupusQoL. Furthermore, the magnitudes of the R^2^ ranged from 0.023 to 0.205 in SLEDAI-2K and 0.021 to 0.216 in SLE-DAS support the view that HRQoL is a different entity from disease activity. Reduced disease activity as a result of treatment may not correlate with improved HRQoL because of the side effects of the medication [27]. Therefore, both of these entities need to be measured for a more complete clinical picture.

The agreement between SLEDAI-2K and SLE-DAS was evaluated using Spearman’s correlation coefficient. In the original SLE-DAS study, SLE-DAS was shown to be strongly correlated with SLEDAI-2K measured at the last follow-up visit of the external validation cohort, with a ρ = 0.94 [14]. In our study, a ρ of 0.78 was observed between SLEDAI-2K and SLE-DAS, which is similar to the 0.70 in a study of 41 Indian patients with lupus nephritis [17]. The low correlation could be attributed to a difference in the distribution of the disease activity between the studies. In a study of 227 Latin American patients with SLE, the authors pointed out that the correlation appeared to depend on the level of the disease activity, with a stronger correlation observed in patients with quiescence or low disease activity [15].

Regarding the associations with various domains of the LupusQoL, SLEDAI-2K and SLE-DAS were similar. When adjusting for sex and age interval, both SLEDAI-2K and SLE-DAS were significantly and inversely associated with all eight domains of LupusQoL. In terms of the magnitude of the standardized beta coefficients of SLEDAI-2K and SLE-DAS, while their rankings were not identical, they were in general agreement. Burden to others, pain, and physical health were the top three domains, whereas emotional health, body image, and planning were the bottom three domains. Several previous studies on patients with SLE from different cultural and ethnic groups have shown varying degrees of association between disease activity and HRQoL. Some studies showed that all the domains were significantly associated with active disease status, whereas some did not. In a study assessing the psychometric properties of LupusQoL in 208 Chinese patients with SLE, the Chinese version of LupusQoL could discriminate patients with active disease activity, defined as a SLEDAI score >4, in all domains except for body image [24]. In addition, a study on 132 Turkish patients with SLE found that all domains except planning of the Turkish version of LupusQoL were able to discriminate between active and inactive SLE groups [28]. Moreover, a study on 78 Iranian patients with SLE showed that active disease, assessed by SLEDAI-2K, was significantly associated with planning, emotional health, and body image domains of the Persian version of the LupusQoL [29]. Furthermore, a cohort study of 182 French patients with SLE showed that the French version of LupusQoL was significantly lower only for physical health, pain, and intimate relationship in patients with SLEDAI >4 [30]. Conversely, no significant differences in any domains of an Argentine version of LupusQoL were observed between 147 patients with a SLEDAI score of <4 and ≥4 [31]. The heterogeneity of the findings from the abovementioned studies might be explained by differences in ethnic composition, cultural setting, and healthcare infrastructure, which could affect the perception of HRQoL in patients with SLE [32].

Our study has some limitations that deserve mention. First, our patients were enrolled from our outpatient clinic, and therefore, the disease activities were relatively mild. Correlations of SLEDAI-2k and SLE-DAS and LupusQoL in patients with more severe disease activity should be investigated in future studies. Second, we did not measure other variables that might potentially affect HRQoL. Nevertheless, we adjusted the association between the two indexes and HRQoL for age and sex, which are likely to be the two most notable potential confounders of the associations. Despite these limitations, to the best of our knowledge, this is the first study to compare the association of HRQoL between SLEDAI-2k and SLE-DAS. The large sample size is also a strength of this study.

In conclusion, findings from this study showed that there were no clear differences in the use of SLE-DAS over SLEDAI-2K in assessing various domains of HRQoL in patients with SLE. We suggest that, in this aspect, both SLEDAI-2K and SLE-DAS are comparable in their associations with disease activity in patients with SLE.

## Figures and Tables

**Figure 1 jcm-10-02137-f001:**
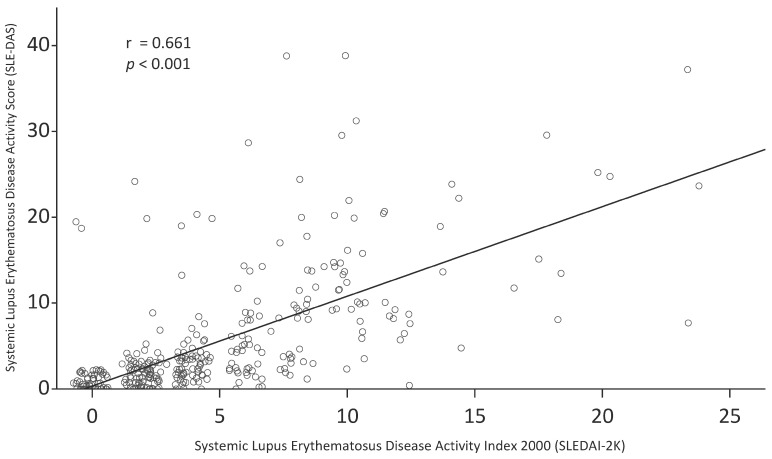
Scatter plot of Systemic Lupus Erythematosus Disease Activity Index 2000 (SLEDAI-2K) and Systemic Lupus Erythematosus Disease Activity Score (SLE-DAS).

**Table 1 jcm-10-02137-t001:** Demographic data of patients with systemic lupus erythematosus (*N* = 333).

Variable	*n*	(%)
Sex		
female	301	(90.4)
male	32	(9.6)
Age interval (years)		
20–29	40	(12.0)
30–39	94	(28.2)
40–49	78	(23.4)
50–59	64	(19.2)
≥60	57	(17.1)
Body mass index (kg/m^2^)		
normal (≥18.5 and <24.0)	179	(53.8)
other	154	(46.2)
Educational level		
high school or below	165	(49.5)
college or above	168	(50.5)
Marital status		
single	111	(33.3)
married, widowed, divorced	222	(66.7)
Change job due to SLE		
no	237	(71.2)
yes	96	(28.8)
Employment status		
unemployed	119	(35.7)
employed	214	(64.3)
Self-perceived health status		
good or very good	90	(27.0)
average	189	(56.8)
poor or very poor	54	(16.2)
Disease duration, years		
≤9	121	(36.3)
>9	212	(63.7)
Age at diagnosis of SLE, years		
<29	177	(53.2)
≥30	156	(46.8)
Alcohol use		
no	255	(76.6)
yes	78	(23.4)
Smoking		
no	301	(90.4)
yes	32	(9.6)
Betel nut chewing		
no	326	(97.9)
yes/ever	7	(2.1)
Regular exercise		
no	57	(17.1)
yes	276	(82.9)
Sleep duration, hours		
0–7	268	(80.5)
≥8	65	(19.5)
Low complement level	184	(55.3)
Increased anti-dsDNA antibody titer	117	(35.1)
Thrombocytopenia (<100,000/mm^3^)	12	(3.6)
Leukopenia (<3000/mm^3^)	17	(5.1)
Anemia	138	(41.4)
Raynaud’s phenomenon	205	(61.6)
Photosensitivity	172	(51.7)
Sjögren’s syndrome	93	(27.9)
Arthritis	72	(21.6)
Renal involvement	47	(14.1)

**Table 2 jcm-10-02137-t002:** Systemic Lupus Erythematosus Disease Activity Score (SLE-DAS), Systemic Lupus Erythematosus Disease Activity Index 2000 (SLEDAI-2K), and Lupus Quality of Life (LupusQoL) of patients with systemic lupus erythematosus (*N* = 333).

Variable	Mean	Standard Deviation	Median	Interquartile Range
SLEDAI-2K	4.87	(4.42)	4.00	(2.00, 7.50)
SLE-DAS	5.43	(6.97)	2.08	(1.12, 8.24)
Domain of LupusQoL				
Physical health	81.2	(20.0)	87.5	(75.0, 96.9)
Emotional health	83.0	(20.1)	87.5	(75.0, 100.0)
Body image	82.4	(23.4)	90.0	(70.0, 100.0)
Pain	80.0	(26.9)	91.7	(75.0, 100.0)
Planning	81.2	(26.0)	91.7	(75.0, 100.0)
Fatigue	72.0	(23.8)	75.0	(56.2, 93.8)
Intimate relationships	73.8	(33.4)	87.5	(62.5, 100.0)
Burden to others	72.1	(30.3)	75.00	(58.3, 100.0)

3.0% (*N* = 10) in the body image domain and 22.2% (*N* = 74) in the intimate relationships of the responses were missing because items were reported as not applicable by the patients.

**Table 3 jcm-10-02137-t003:** Linear regression analyses of the eight domains of Lupus Quality of Life (LupusQoL) with the Systemic Lupus Erythematosus Disease Activity Index 2000 (SLEDAI-2K) in patients with systemic lupus erythematosus.

Domain of LupusQoL	Simple Linear Regression Analysis				Linear Regression Analysis Adjusted for Sex and Age Interval			
	β	(95% CI)	Std. β	*p*	β	(95% CI)	Std. β	*p*
Physical health	−0.459	(−0.945, 0.028)	−0.101	0.065	−0.871	(−1.347, −0.394)	−0.192	<0.001
Emotional health	−0.523	(−1.012, −0.034)	−0.115	0.036	−0.670	(−1.181, −0.159)	−0.147	0.010
Body image	−0.628	(−1.205, −0.051)	−0.119	0.033	−0.673	(−1.276, −0.069)	−0.127	0.029
Pain	−0.715	(−1.370, −0.059)	−0.117	0.033	−1.196	(−1.859, −0.532)	−0.196	<0.001
Planning	−0.627	(−1.262, 0.007)	−0.106	0.053	−0.728	(−1.392, −0.063)	−0.123	0.032
Fatigue	−0.866	(−1.441, −0.290)	−0.160	0.003	−0.997	(−1.598, −0.397)	−0.185	0.001
Intimate relationships	−0.228	(−1.156, 0.699)	−0.030	0.628	−1.297	(−2.175, −0.419)	−0.172	0.004
Burden to others	−1.663	(−2.383, −0.944)	−0.243	<0.001	−1.633	(−2.389, −0.877)	−0.238	<0.001

CI: confidence interval; std: standardized. 3.0% (*N* = 10) in the body image domain and 22.2% (*N* = 74) in the intimate relationships of the responses were missing because these items were reported as not applicable.

**Table 4 jcm-10-02137-t004:** Linear regression analyses of the eight domains of Lupus Quality of Life (LupusQoL) with the Systemic Lupus Erythematosus Disease Activity Score (SLE-DAS) in patients with systemic lupus erythematosus.

Domain of LupusQoL	Simple Linear Regression Analysis				Linear Regression Analysis Adjusted for Sex and Age Interval			
	β	(95% CI)	Std. β	*p*	β	(95% CI)	Std. β	*p*
Physical health	−0.469	(−0.775, −0.164)	−0.164	0.003	−0.623	(−0.913, −0.332)	−0.217	<0.001
Emotional health	−0.423	(−0.731, −0.115)	−0.147	0.007	−0.469	(−0.782, −0.156)	−0.163	0.003
Body image	−0.356	(−0.720, 0.007)	−0.107	0.055	−0.385	(−0.755, −0.015)	−0.115	0.042
Pain	−0.642	(−1.054, −0.230)	−0.166	0.002	−0.784	(−1.191, −0.377)	−0.203	<0.001
Planning	−0.381	(−0.783, 0.021)	−0.102	0.063	−0.429	(−0.838, −0.021)	−0.115	0.039
Fatigue	−0.594	(−0.957, −0.230)	−0.174	0.001	−0.638	(−1.006, −0.269)	−0.187	<0.001
Intimate relationships	−0.664	(−1.230, −0.098)	−0.041	0.022	−0.915	(−1.434, −0.396)	−0.197	0.001
Burden to others	−0.943	(−1.401, −0.485)	−0.217	<0.001	−0.936	(−1.402, −0.470)	−0.216	<0.001

CI: confidence interval; std: standardized. 3.0% (*N* = 10) in the body image domain and 22.2% (*N* = 74) in the intimate relationships of the responses were missing because these items were reported as not applicable.

**Table 5 jcm-10-02137-t005:** Regression model accuracy metrics of the eight domains of Lupus Quality of Life (LupusQoL) with Systemic Lupus Erythematosus Disease Activity Score (SLE-DAS) and Systemic Lupus Erythematosus Disease Activity Index 2000 (SLEDAI-2K) adjusted for age and sex.

Domain of LupusQoL	SLE-DAS					SLEDAI-2K					*p*	
	MAE	RMSE	AIC	BIC	R^2^	MAE	RMSE	AIC	BIC	R^2^	MAE	RMSE
Physical health	13.31	18.28	2904.7	2923.8	0.159	13.47	18.41	2905.9	2925.0	0.147	0.370	0.578
Emotional health	14.71	19.68	2942.9	2961.9	0.037	14.68	19.74	2944.7	2963.8	0.032	0.859	0.722
Body image	17.58	23.15	2958.7	2977.6	0.021	17.60	23.13	2957.7	2976.6	0.023	0.899	0.864
Pain	19.29	25.60	3116.9	3135.9	0.094	19.51	25.67	3118.0	3137.0	0.089	0.322	0.758
Planning	19.34	25.70	3120.2	3139.2	0.024	19.31	25.68	3119.1	3138.2	0.025	0.829	0.909
Fatigue	18.89	23.18	3051.8	3070.8	0.050	18.89	23.21	3052.0	3071.0	0.048	0.965	0.865
Intimate relationships	23.27	29.52	2504.9	2522.7	0.216	23.43	29.72	2506.7	2524.5	0.205	0.583	0.548
Burden to others	23.48	29.34	3208.2	3227.2	0.058	23.33	29.23	3205.8	3224.9	0.064	0.580	0.702

AIC: Akaike Information Criterion; BIC: Bayesian Information Criterion; MAE: Mean Absolute Error; RMSE: Root Mean Square Error. 3.0% (*N* = 10) in the body image domain and 22.2% (*N* = 74) in the intimate relationships of the responses were missing because these items were reported as not applicable.

**Table 6 jcm-10-02137-t006:** Regression model accuracy metrics of the eight domains of Lupus Quality of Life (LupusQoL) with Systemic Lupus Erythematosus Disease Activity Score (SLE-DAS) and Systemic Lupus Erythematosus Disease Activity Index 2000 (SLEDAI-2K) adjusted for age and sex in SLE patients with renal involvement (*N* = 47).

Domain of LupusQoL	SLE-DAS					SLEDAI-2K					*p*	
	MAE	RMSE	AIC	BIC	R^2^	MAE	RMSE	AIC	BIC	R^2^	MAE	RMSE
Physical health	12.17	17.81	419.4	428.7	0.191	10.70	15.76	408.3	417.5	0.367	0.252	0.178
Emotional health	12.84	17.08	419.4	428.6	0.198	12.68	16.90	418.7	428.0	0.216	0.765	0.594
Body image	18.04	25.05	448.4	457.7	0.091	18.17	24.22	446.5	455.8	0.150	0.890	0.428
Pain	16.43	22.47	442.2	451.4	0.290	16.59	21.50	437.9	447.2	0.350	0.900	0.436
Planning	19.12	25.82	450.6	459.9	0.100	18.28	24.01	443.0	452.2	0.222	0.547	0.144
Fatigue	16.97	20.60	434.5	443.7	0.155	16.64	20.53	434.4	443.6	0.161	0.575	0.883
Intimate relationships (*n* = 37)	21.17	26.61	363.4	371.4	0.372	21.32	27.13	364.1	372.1	0.347	0.867	0.654
Burden to others	23.18	28.65	466.1	475.3	0.191	22.82	27.96	462.4	471.7	0.230	0.790	0.602

AIC: Akaike Information Criterion; BIC: Bayesian Information Criterion; MAE: Mean Absolute Error; RMSE: Root Mean Square Error; 22.3% (*N* = 10) in intimate relationships of the responses were missing because these items were reported as not applicable.

**Table 7 jcm-10-02137-t007:** Regression model accuracy metrics of the eight domains of Lupus Quality of Life (LupusQoL) with Systemic Lupus Erythematosus Disease Activity Score (SLE-DAS) and Systemic Lupus Erythematosus Disease Activity Index 2000 (SLEDAI-2K) adjusted for age and sex SLE patients without renal involvement (*N* = 286).

Domain of LupusQoL	SLE-DAS					SLEDAI-2K					*p*	
	MAE	RMSE	AIC	BIC	R^2^	MAE	RMSE	AIC	BIC	R^2^	MAE	RMSE
Physical health	13.19	18.00	2487.8	2506.1	0.186	13.51	18.28	2492.6	2510.8	0.160	0.107	0.331
Emotional health	14.92	19.83	2532.2	2550.5	0.037	14.88	19.92	2535.2	2553.4	0.027	0.772	0.645
Body image	17.38	22.69	2517.6	2535.7	0.013	17.37	22.69	2517.4	2535.5	0.013	0.879	0.996
Pain	19.56	25.79	2682.0	2700.2	0.082	19.78	25.89	2683.9	2702.2	0.075	0.305	0.659
Planning	19.18	25.55	2677.3	2695.6	0.018	19.20	25.59	2677.7	2696.0	0.015	0.865	0.720
Fatigue	19.01	23.35	2625.1	2643.4	0.052	19.05	23.39	2625.7	2644.0	0.049	0.878	0.880
Intimate relationships	23.49	29.62	2149.4	2166.4	0.208	23.67	29.84	2151.5	2168.5	0.197	0.607	0.580
Burden to others	23.09	29.15	2751.3	2769.6	0.043	23.07	29.09	2750.8	2769.1	0.047	0.938	0.837

AIC: Akaike Information Criterion; BIC: Bayesian Information Criterion; MAE: Mean Absolute Error; RMSE: Root Mean Square Error; 3.5% (*N* = 10) in the body image domain and 22.4% (*N* = 64) in the intimate relationships of the responses were missing because these items were reported as not applicable.

## Data Availability

The data presented in this study are available on request from the corresponding author.

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
