# Peer review of "A Comparison of the Correlation of Systemic Lupus Erythematosus Disease Activity Index 2000 (SLEDAI-2K) and Systemic Lupus Erythematosus Disease Activity Score (SLE-DAS) with Health-Related Quality of Life"

_jcm, 2021, doi:10.3390/jcm10102137_

Round 1

Reviewer 1 Report

It is always important to evaluate performances of newly developed scoring system in different cultural and ethnic groups. In this manuscript, the authors showed that lupus erythematosus disease activity score (SLE-DAS), the newly developed scoring system for lupus patients, could be applied to their patients. The authors also showed there were no clear differences in the use of SLE-DAS over systemic lupus erythematosus disease activity index 2000, the former scoring system, in assessing health-related quality of life in their patients. This manuscript may give a useful information to rheumatologists in evaluating their patients’ disease activity with SLE-DAS.

Author Response

We thank the reviewer for taking the time and effort necessary to review our manuscript.

Reviewer 2 Report

The paper is well written and the methods are clearly described.

The investigated population of SLE patients seems to be heterogeneous as far as the disease activity is concerned. 

The authors should describe in detail the main reasons for disease activity. Different domains in LupusQoL might be influenced by different clinical phenotypes (arthritis vs nephritis or neurolupus). The authors adjusted the disease activity data only for sex and age, without taking into account the clinical phenotypes.

Are SLEDAI-2K and SLE-DAS comparable in assessing various domains of HRQoL regardless of the clinical phenotypes?

Author Response

Reviewer 2 Comment 1:

The paper is well written and the methods are clearly described.

The investigated population of SLE patients seems to be heterogeneous as far as the disease activity is concerned.

The authors should describe in detail the main reasons for disease activity. Different domains in LupusQoL might be influenced by different clinical phenotypes (arthritis vs nephritis or neurolupus). The authors adjusted the disease activity data only for sex and age, without taking into account the clinical phenotypes.

Response to Reviewer 2, Comment 1:

We thank the reviewer for the invaluable suggestion. Different domains in LupusQoL indeed might be influenced by different clinical phenotypes, such as arthritis. The reason for not including the adjustment of these clinical phenotypes in Table 5 is because the presence of these phenotypes was already included in the scoring of SLEDAI-2K and SLE-DAS.

-----------------------------------------------------------------------------------------

Reviewer 2, Comment 2:

Are SLEDAI-2K and SLE-DAS comparable in assessing various domains of HRQoL regardless of the clinical phenotypes?

Response to Reviewer 2, Comment 2:

Following the suggestion by the reviewer, we conducted further analyses comparing the association of of each of the eight domains of LupusQoL with SLE-DAS and SLEDAI-2K in SLE patients with or without renal involvement (new Table 6 and 7, respectively). The magnitudes of MAE, RMSE, AIC, BIC, and R2 were comparable between SLEDAI-2K and SLE-DAS. In addition, MAE and RMSE obtained from SLEDAI-2K and SLE-DAS were not significantly different for all eight domains of LupusQoL in patients with SLE with or without renal involvement.

Reviewer 3 Report

This is a cross-sectional study exploring the correlation between SLEDAI-2K and SLE-DAS and their correlations with components of the Lupus QoL, a SLE-customized QoL index. I have some concerns regarding the actual design of the study and the results, which I think merit consideration.

1) Overall concept: The idea to compare two indices of disease activity in SLE is valid, in order to further validate the new SLE-DAS. Examination of their association with QoL is also useful. However the two indices have not been designed to assess QoL, they have been designed to assess disease activity! Thus, their "performance" cannot be tested in this way. Instead, it would be tested if they correlated the two indices with eg. changes in treatment. The authors are thus asked to rephrase all respective parts of the document, by discussing correlations between indices and QoL, but not referring to performance or anything similar. As an example, in the Abstract Conclusion: "There were no clear differences in the use of SLE-DAS over SLEDAI-2K in assessing HRQoL in patients with SLE. We suggest that, in this aspect, both SLEDAI-2K and SLE-DAS are effective tools for measuring disease activity in patients with SLE.", both sentences need rephrasing. The indices are not designed to assess QoL, and the effectiveness in measuring disease activity needs correation with a different parameter. Similar changes are needed throughout the manuscript.

Abstract: In Methods, some parts of the analyses performed should be described briefly.

Abstract, Results and Results in general: Mean (SD) of SLEDAI and SLE-DAS obviously show a non-normal distribution. Clearly, median (IQR) should be used instead.

Methods-Results: To assess correlation between disease indices and components of Lupus-QoL, authors have adjusted only for sex and age. But, QoL is dependent on multiple parameters, beyond disease activity (as the authors themselves acknowledge at some point in their manuscript). This is a major limitation of the study.

Results, Figure 1: The dots in the scattter plot seem much less than 333. Shouldn't all individual patients be represented therein? Please clarify.

Results: Clinical characteristics of this large patient sample are largedly undescribed. Table 1 mainly focuses on demographics and habits. Some clinical characteristics, eg. percentage of LN or other clinical manifestations should be added. With such a large patient sample , it would be interesting to check also correlation of SLEDAI with SLE-DAS in different subsets of patients (eg LN vs no-LN, or others)

Minor: Methods and Results: Why was betel nut chewing in particular assessed?

Author Response

Reviewer 3, Comment 1:

 This is a cross-sectional study exploring the correlation between SLEDAI-2K and SLE-DAS and their correlations with components of the Lupus QoL, a SLE-customized QoL index. I have some concerns regarding the actual design of the study and the results, which I think merit consideration.

1) Overall concept: The idea to compare two indices of disease activity in SLE is valid, in order to further validate the new SLE-DAS. Examination of their association with QoL is also useful. However the two indices have not been designed to assess QoL, they have been designed to assess disease activity! Thus, their "performance" cannot be tested in this way. Instead, it would be tested if they correlated the two indices with eg. changes in treatment. The authors are thus asked to rephrase all respective parts of the document, by discussing correlations between indices and QoL, but not referring to performance or anything similar. As an example, in the Abstract Conclusion: "There were no clear differences in the use of SLE-DAS over SLEDAI-2K in assessing HRQoL in patients with SLE. We suggest that, in this aspect, both SLEDAI-2K and SLE-DAS are effective tools for measuring disease activity in patients with SLE.", both sentences need rephrasing. The indices are not designed to assess QoL, and the effectiveness in measuring disease activity needs correation with a different parameter. Similar changes are needed throughout the manuscript.

Response to Reviewer 3, Comment 1:

We thank the reviewer for the comment. We have rephrased all relevant sentences throughout the manuscript.

We have also changed the title of the manuscript to “A comparison of the correlation between systemic lupus erythematosus disease activity index 2000 (SLEDAI-2K) and systemic lupus erythematosus disease activity score (SLE-DAS) with health-related quality of life”.

--------------------------------------------------------------------------------

Reviewer 3, Comment 2:

Abstract: In Methods, some parts of the analyses performed should be described briefly.

Response to Reviewer 3, Comment 2:

The summary statistics presented in Table 1 and 2 are described in the Statistical analysis section as “Continuous variables were summarized as mean with standard deviation (SD) and median with interquartile range, as appropriate. Categorical variables were presented as frequencies and percentages.”

The results presented in Table 3 and 4 are described as “Separate linear regression analyses for each of the eight domains of LupusQoL were performed with SLEDAI-2K and SLE-DAS as independent variables. Because sex differences were observed in HRQoL in patients with SLE [25], linear regression models were fitted with and without adjusting for sex and age interval.”.

The results shown in Table 5 to 7 are described as “The performance of SLEDAI-2K and SLE-DAS was compared by evaluating their associations with LupusQoL using five regression model accuracy metrics, including mean absolute error (MAE), root mean square error (RMSE), Akaike Information Cri-terion (AIC), Bayesian Information Criterion (BIC), and coefficient of determination (R2).” The methods in the calculation of these indices are described in reference 26.

The p values in Table 5 to 7 comparing MAE and RMSE between SLE-DAS and SLEDAI-2K were obtained using paired t-test.

"In addition, the correlation between SLEDAI-2K and SLE-DAS was determined using Pearson’s correlation coefficient and Spearman’s rank correlation coefficient." The correlation coefficient shown in Figure 1 was Pearson’s correlation coefficient.

--------------------------------------------------------------------------------

Reviewer 3, Comment 3:

Abstract, Results and Results in general: Mean (SD) of SLEDAI and SLE-DAS obviously show a non-normal distribution. Clearly, median (IQR) should be used instead.

Response to Reviewer 3, Comment 3:

We have changed the summary statistics to median IQR.

--------------------------------------------------------------------------------

Reviewer 3, Comment 4:

Methods-Results: To assess correlation between disease indices and components of Lupus-QoL, authors have adjusted only for sex and age. But, QoL is dependent on multiple parameters, beyond disease activity (as the authors themselves acknowledge at some point in their manuscript). This is a major limitation of the study.

Response to Reviewer 3, Comment 4:

We thank the reviewer for the comment. We agree with the reviewer that QoL can depend on various parameters, which we had not measured in our study. We have added the following sentences in the limitation section: “Second, we did not measure other variables that might potentially affect HRQoL. Nevertheless, we have adjusted the association between the two indexes and HRQoL for age and sex, which are likely to be the two most notable potential confounders of the associations.”

--------------------------------------------------------------------------------

Reviewer 3, Comment 5:

Results, Figure 1: The dots in the scatter plot seem much less than 333. Shouldn't all individual patients be represented therein? Please clarify.

Response to Reviewer 3, Comment 5:

The reason for that is more than one patient had the same SLEDAI-2K and SLE-DAS and therefore, their data points were overlapped. We have re-plotted the scatterplot with jitter and non-filled data point symbol to overcome this issue in our revised Figure 1.

--------------------------------------------------------------------------------

Reviewer 3, Comment 6:

Results: Clinical characteristics of this large patient sample are largedly undescribed. Table 1 mainly focuses on demographics and habits. Some clinical characteristics, eg. percentage of LN or other clinical manifestations should be added. With such a large patient sample, it would be interesting to check also correlation of SLEDAI with SLE-DAS in different subsets of patients (eg LN vs no-LN, or others)

Response to Reviewer 3, Comment 6:

We have added several clinical characteristics of SLE in our revised Table 1. In addition, we have added analyses comparing the association of each of the eight domains of LupusQoL with SLE-DAS and SLEDAI-2K in SLE patients with or without renal involvement (new Table 6 and 7, respectively). The magnitudes of MAE, RMSE, AIC, BIC, and R2 were comparable between SLEDAI-2K and SLE-DAS. In addition, MAE and RMSE obtained from SLEDAI-2K and SLE-DAS were not significantly different for all eight domains of LupusQoL in patients with SLE with or without renal involvement.

--------------------------------------------------------------------------------

Reviewer 3, Comment 7:

Minor: Methods and Results: Why was betel nut chewing in particular assessed?

Response to Reviewer 3, Comment 7:

Betel nut chewing is a widespread habit in Taiwan, especially in its southern and eastern regions. Since the study participants were recruited from a regional hospital in southern Taiwan, and betel nut chewing can be associated some medical comorbidities, we included this variable in our study. Nevertheless, a very low prevalence (2.1%) of betel nut chewing experience was observed in our patients with SLE.

--------------------------------------------------------------------------------

Round 2

Reviewer 2 Report

The quality of the paper has improved after revision.

Reviewer 3 Report

The authors have addressed my comments that could be addressed. No further comments.

This manuscript is a resubmission of an earlier submission. The following is a list of the peer review reports and author responses from that submission.

Round 1

Reviewer 1 Report

Recent some studies have addressed the clinically association of TCM body constitution with various diseases. Authors stated numbers of unbalanced body constitution types are associated with QOL in SLE patients. However, it seems that the analysis of various factors has not been sufficiently conducted.

  1. It is unclear why only women participated in the study.
  2. Since SLE shows a very diverse clinical course and disease state, the effect on QOL could not be assessed easily. That is, the effect of organ involvement and medications on healthy related QOL would be more significant than suggested body constitution in patients with SLE. Nevertheless multiple analysis was performed without the various parameters.
  3. In order to suggest the effect on clinical indicators of TCM body constitution, longitudinal observation study design rather than cross-sectional study seems to be appropriate.

Reviewer 2 Report

The authors present the association of traditional Chinese medicine (TCM) body constitution and different quality of life domains in patients with systemic lupus erythematosus (SLE). Although the results of this study are potentially interesting, the following critiques should be addressed.

1. The authors did not describe the treatments in the manuscript. It seems that the most patients had been receiving some immunosuppressants. It would be more interesting if the authors discuss whether the results in this study present the disease character or the treatments effects.

2. Is there a correlation between the TCM body constitution types and SLEDAI-2K?

3. The authors showed three clusters of unbalanced TCM body constitution in Figure 2.  Is this original in SLE? Is it possible to compare to that from the other rheumatic diseases such as rheumatoid arthritis?